# Glass-Ceramic Foams from Alkali-Activated Vitrified Bottom Ash and Waste Glasses

**Miroslava Hujova [1,\*], Patricia Rabelo Monich [2], Jaroslav Sedlacek [3,4], Miroslav Hnatko [3,4], Jozef Kraxner [1], Dusan Galusek [1,5]** and **Enrico Bernardo [2]**

[1] FunGlass, Alexander Dubcek University of Trencin, 91101 Trencin, Slovakia; jozef.kraxner@tnuni.sk (J.K.); dusan.galusek@tnuni.sk (D.G.)
[2] Dipartmento di Ingegneria Industriale Università degli Studi di Padova, 35112 Padova, Italy; patricia.rabelomonich@phd.unipd.it (P.R.M.); enrico.bernardo@unipd.it (E.B.)
[3] Institute of Inorganic Chemistry, Slovak Academy of Sciences, 81438 Bratislava, Slovakia; Jaroslav.Sedlacek@savba.sk (J.S.); uachmiho@savba.sk (M.H.)
[4] Centre of Excellence for Advanced Materials Application, Slovak Academy of Sciences, 81438 Bratislava, Slovakia
[5] Joint Glass Centre of the IIC SAS, TnUAD, and FChFT STU, 91101 Trencin, Slovakia
[\*] Correspondence: Miroslava.hujova@tnuni.sk

**Abstract:** Both vitrified bottom ashes (VBAs) and waste glasses are forms of inorganic waste material that are widely landfilled, despite having some economic potential. Building on previous studies, we prepared glass-ceramic foams by the combination of VBA with either soda-lime glass (SLG) or borosilicate glass (BSG). Suspensions of fine powders in weakly alkaline solution underwent gelation, followed by frothing at nearly room temperature. Hardened "green" foams were sintered, with concurrent crystallization, at 850–1000 °C. All foams were highly porous (>70%), with mostly open porosity. The glass addition was fundamental in both gelation (promoting the formation of carbonate and silicate hydrated phases) and firing steps. While SLG addition enhanced the viscous flow sintering, without a significant impact on the crystallization of gehlenite, the main crystalline phase from the devitrification of VBA, BSG addition caused a reactive sintering, with remarkable changes in the phase assemblage. The glass addition generally also allowed lower sintering temperatures and yielded products with excellent crushing strength. However, only specific conditions resulted in the complete immobilization of pollutants (e.g., $Cr^{3+}$ ions).

**Keywords:** alkali activation; sintering; glass foams; chemical durability

## 1. Introduction

The impacts of human activities and the consequential waste production have become one of the most discussed topics in the 21st century. Currently, waste management aims to tackle landfilling through recycling and incineration. However, both of these options to some degree yield byproducts that are still destined for landfill.

The reuse of cullet by remelting into new glass products is an excellent example of imperfect recycling. Within the EU alone, in 2014, the production of glass waste was close to 18.5 million tonnes, but it is estimated that ~25% of this waste remained unrecovered [1]. Even the adoption of the most sophisticated recycling systems does not imply the full recovery of soda-lime glass (SLG), especially concerning finely powdered fractions, enriched in contaminants, which are landfilled [2], along with specialty glasses, like opal [3], sheet [4], LCD [5], or pharmaceutical [6] glasses, or glass fibers [3].

Mixed municipal waste is, by definition, unrecyclable. It is the most abundant waste category due to ever-growing consumption, and in this case, incineration is single-handedly favored [7].

A fundamental byproduct of incineration consists of bottom ash, which is 20–25% of the waste input volume. Such ash, 90% of which is composed of minerals and glass [8,9], is in turn subjected to secondary thermal treatments, mainly aimed at metal recovery, with formation of metal-free vitrified bottom ash (VBA) [10]. It is worth mentioning that in Europe, 10 countries are directly landfilling VBA, and only 5 countries utilize it for more than 80% of the produced amount [11].

If utilized, VBA is used for civil engineering projects (i.e., construction of highways) [11], with only limited valorization. However, in recent years, new business has emerged in the post-treatment technologies of bottom ashes and in the production of new materials from the perspective of "end-of-waste" philosophy [12].

In the past, VBA was investigated in products such as tiles and bricks [1]. More recent studies have been concerned with reuse of cellular glass-based materials, which are particularly attractive from an environmental perspective. Energy and emissions savings in reusing waste-derived glasses (instead of glasses from conventional raw materials) are accompanied by savings in their long service life. Glass-based foams are thermal and acoustic insulators with much greater durability than polymer foams [13]. Baino et al. [13] successfully prepared ceramic foams by pressing and sintering VBA mixed with foaming agents, such as borax and calcite. Romero et al. [14], on the contrary, could develop foams from VBA without foaming additives. Foams were prepared by the intensive mechanical stirring of glass suspensions in weakly alkaline solutions, followed by curing at near room temperatures, and sintering with concurrent crystallization. The alkaline activation was not intended to determine an extensive dissolution (as occurring in inorganic polymers, developed by means of solutions with higher alkalinity and including synthetic components such as alkali silicates [15]). It was specifically aimed at the formation of gels at the surface of glass powders, in turn enhancing the viscosity of suspensions and stabilizing the cellular structures developed upon stirring [16].

Due to its origin, VBA might contain hazardous metals (Cr, Ni, Pb, Hg, etc.) and corrosive pollutants ($Cl^-$, $F^-$, $SO_3^{2-}$), so any upcycling product must also be studied from the aspect of chemical stability [17,18]. Studies on VBA-derived glass-ceramic foams have yielded generally safe materials, that is, with very low leaching values [10,13,14,19]. In any case, any processes should be verified according to the variability of chemical compositions.

In this study, we focused on the production of glass-ceramic foams from relatively silica-poor VBA, exploiting the above-mentioned process of weak alkali activation and mechanical foaming, before viscous flow sintering. More precisely, we referred to a glass with $SiO_2$ well below 50 wt%, featuring high content of CaO and $Al_2O_3$. The feasibility of glass with this composition has already been reported in previous publications, in different contexts [20,21]. Both gelation and sintering were conditioned by mixing VBA with the two most common waste glasses: (i) residues from soda-lime glass (SLG) recycling; and (ii) medicinal borosilicate glass (BSG). Without the glass addition, VBA-based glass-ceramic foams demonstrated the crystallization of gehlenite, which was demonstrated by its poor mechanical properties. The glass addition caused changes in the phase assemblage and enabled lower sintering temperatures. The research also suggests that only specified conditions lead to the satisfactory compromise between firing temperature (to be decreased), and mechanical and chemical resistance of the final material (to be increased).

## 2. Materials and Methods

The starting materials consisted of vitrified bottom ash (VBA; mean particle size <75 μm), soda-lime glass (SLG; courtesy of SASIL Srl, Biella, Italy; mean particle size <30 μm) and borosilicate glass (BSG, courtesy of Nuova OMPI, Padova, Italy; mean particle size <75 μm). VBA was derived from the processing of incinerator bottom ash by means of DC plasma reactor [22] at 1400 °C (courtesy of SAV Bratislava, Slovakia). The chemical compositions, obtained via XRF, are provided in Table 1. VBA was also subjected to thermal analysis (differential scanning calorimetry (DSC), Mettler Toledo TGA/DSC 3+, STARe System, Columbus, OH, USA), in order to determine the glass transition temperature ($T_g$) and the crystallization temperature ($T_c$).

**Table 1.** Chemical composition (in wt %) of waste streams.

|  | Vitrified Bottom Ash (VBA) | Soda-Lime Glass (SLG) | Borosilicate Glass (BSG) |
|---|---|---|---|
| $SiO_2$ | 32.1 | 71.9 | 72 |
| $B_2O_3$ | - | - | 12 |
| CaO | 33.5 | 7.5 | 1 |
| $Na_2O$ | - | 14.3 | 6 |
| MgO | 4.3 | 4.0 | - |
| $Al_2O_3$ | 27.5 | 1.2 | 7 |
| $Fe_2O_3$ | 0.3 | - | - |
| $K_2O$ | 0.2 | - | 2 |
| $TiO_2$ | 0.6 | - | - |
| Residues | 1.6 | 1.1 | - |

Powdered VBA, the main component, was suspended in 2.5 M NaOH aqueous solution, either pure or mixed with SLG or BSG in a 70/30 weight ratio, under constant mechanical stirring (500 rpm), for 2 h. The overall solid content of the suspensions was kept constant at 70 wt %. Afterwards, 4 wt % of surfactant (Triton X-100, (polyoxyethylene octyl phenyl ether, $C_{14}H_{22}O(C_2H_4O)_n$, n = 9–10), Sigma–Aldrich, Gillingham, UK) was added to the suspensions. The suspensions were finally foamed by means of intensive mechanical stirring (2200 rpm), poured into polystyrene molds, and left at 40 °C for 48 h [14].

The foams that developed at nearly room temperature were fired at 850–1000 °C; the heating rate was 10 °C min$^{-1}$, whereas the holding time at the maximum temperature was 1 h. The cooling rate was restricted to 5 °C min$^{-1}$ to avoid cracking of samples. The glass-ceramic samples before and after firing were subjected to: phase analysis via X-ray diffraction (XRD; Bruker D8 Advance, Cu $K_\alpha$ radiation 0.15418 nm, Karlsruhe, Germany); pycnometry (Micromeritics AccuPyc 1330, Norcross, GA, USA); optical microscopy (AxioCam ERc 5 second microscope camera, Carl Zeiss Microscopy, Thornwood, NY, USA); and scanning electron microscopy (SEM FEI Quanta 200 ESEM, Eindhoven, The Netherlands). The phase identification from X-ray diffraction patterns was performed through High Score Plus (PANanalytical B.V, Almelo, The Netherlands).

Representative samples of approximately 10 mm × 10 mm × 10 mm were cut from the main specimens and subjected to compression tests (Quasar 25, Galdabini, Cardano, Italy), operating with a crosshead speed of 1 mm min$^{-1}$. Each data point represents an average value obtained by testing ten specimens.

In order to assess the chemical stability of prepared samples, 1 g of crushed sample (pieces smaller than 10 mm) was placed into a polypropylene test-tube with 10 g of deionized water, in order to obtain a solid/liquid ratio of 1/10 w/w. The suspensions were stirred for 24 h, after which, clear solutions were obtained via decantation, centrifugation, and filtering. Leachates were analyzed via optical emission spectroscopy with induction plasma (ICP-OES, Agilent 7500 cx, Santa Clara, CA, USA).

## 3. Results and Discussion

The VBA demonstrated different behavior from results obtained in previous studies [10,14]. In fact, "green" foams from alkali activation and mechanical foaming, after drying, were extremely brittle. This could be interpreted as the consequence of limited gel formation at the surface of VBA particles, upon attack in weakly alkaline solution. No transformation could be inferred from FTIR spectroscopy (not reported, for the sake of brevity) or diffraction analysis, as shown in Figure 1a. VBA appears to be fully amorphous, with a distinct halo centered at 2θ~30°, except for traces attributed to magnesium silicate (olivine, $Mg_2SiO_4$, PDF#87-2039) and magnesium aluminate (spinel, $(Mg_{0.75}Al_{0.25})(Al_{0.875}Mg_{0.125})_2O_4$, PDF#86-0088).

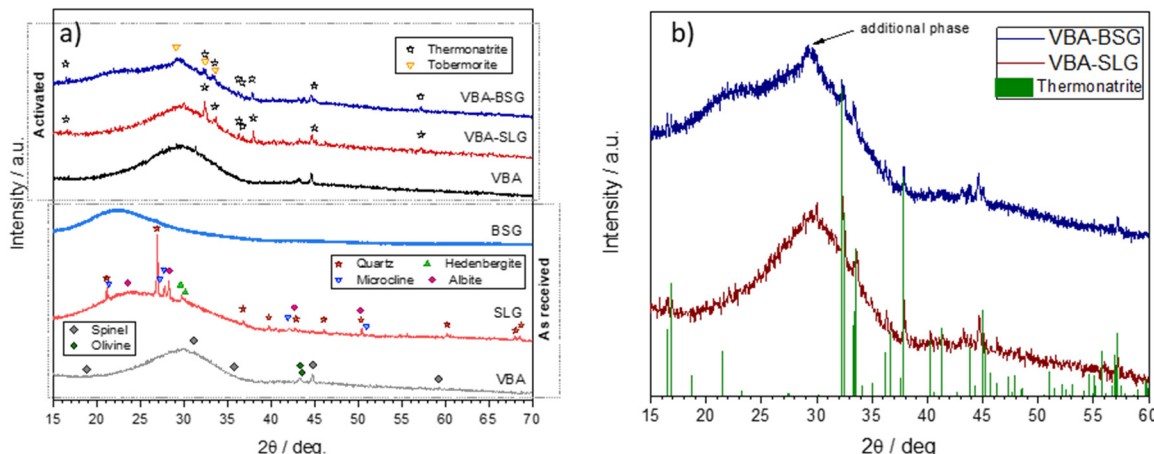

**Figure 1.** XRD analysis of starting materials and foams formed after alkali activation: (**a**) overall comparison; (**b**) detailed view of the patterns of the products of activation of VBA mixed with SLG and BSG glasses.

Stronger foams could be obtained, already in the "green" state, with SLG and BSG addition. As shown in Figure 1a, SLG actually contained some mineral impurities, such as quartz (PDF#79-1906), sodium, and potassium aluminosilicates (albite, $NaAlSi_3O_8$, PDF#09-0466; microcline, $KAlSi_3O_8$, PDF#19-0932) and iron silicate (hedenbergite, $CaFeSi_2O_6$, PDF#71-1501). Such impurities disappeared after alkaline attack; the VBA–SLG mixture underwent hardening according to the formation of a semicrystalline gel, featuring a hydrated sodium carbonate phase (thermonatrite, $Na_2CO_3 \cdot H_2O$, PDF#76-0910), as already found in previous experiments with VBA of different chemical composition [14]. The matching of most peaks in the pattern of activated VBA–SLG mixture with those of thermonatrite is further demonstrated in Figure 1b.

BSG glass was completely amorphous. Interestingly, owing to the different chemical formulation, the amorphous halo was placed at a much lower 2θ than in the case of VBA. Probably due to limited dissolution upon the alkaline attack, the signal persisted in the pattern of activated VBA–BSG mixture (see the shoulder at 2θ~22° in the top pattern in Figure 1a). BSG addition also led to the formation of a semicrystalline gel, with the presence of thermonatrite (Figure 1b). The broad peak at 2θ~30° suggests the formation of an additional phase that is compatible with the formation of hydrated calcium silicate (tobermorite, $Ca_5(Si_6O_{16})(OH)_2$, PDF#89-6458).

The glass addition also had a fundamental impact on the viscous flow sintering. The DSC plot shown in Figure 2 suggests that VBA presents a narrow sintering window, with a first crystallization peak at $T_{cryst,1}$ ~ 940 °C, only 100 °C above the glass transition temperature ($T_g$ ~ 840 °C). Therefore, the viscous flow of softened VBA, upon heating above the $T_g$, could be compromised by the strong viscosity increase associated with crystal precipitation [23]. This explains why the foams from pure VBA did not undergo a significant consolidation upon firing at 950 °C. The foams, which were already extremely brittle in the "green" state, could be crushed even by gentle hand pressure, after firing.

At 950 °C, as shown in Figure 3, VBA predominantly crystallized into gehlenite ($Ca_2Al_2SiO_7$, PDF#79-2421), a relatively silica-poor calcium aluminosilicate, consistent with the low silica content of the starting VBA. For both types of glass addition, the intensity of distinctive gehlenite diffraction lines decreased substantially by more than 30% (amount of glass added). Such a decrease would be expected in the hypothesis of glasses simply providing an inert liquid phase, acting as a glue for VBA particles. SLG and BSG undergo softening well below the $T_g$ of VBA (560 °C for SLG and 720 °C for BSG [24]), and they modified the crystallization sequence substantially.

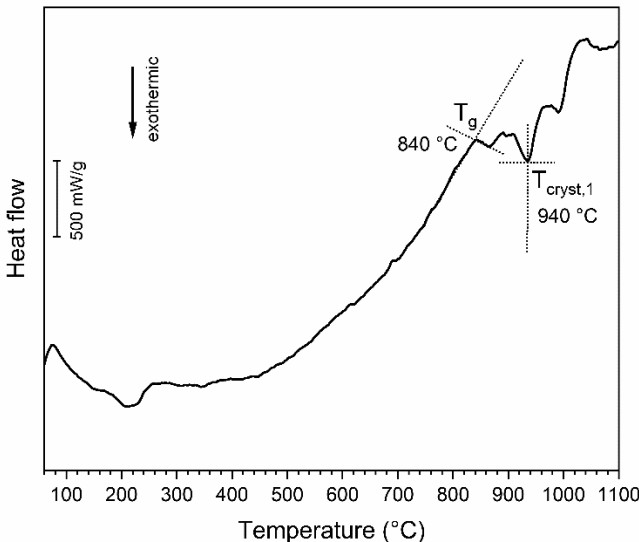

**Figure 2.** DSC plot for VBA powder, showing glass transition temperature ($T_g$) and crystallization temperature ($T_{cryst}$).

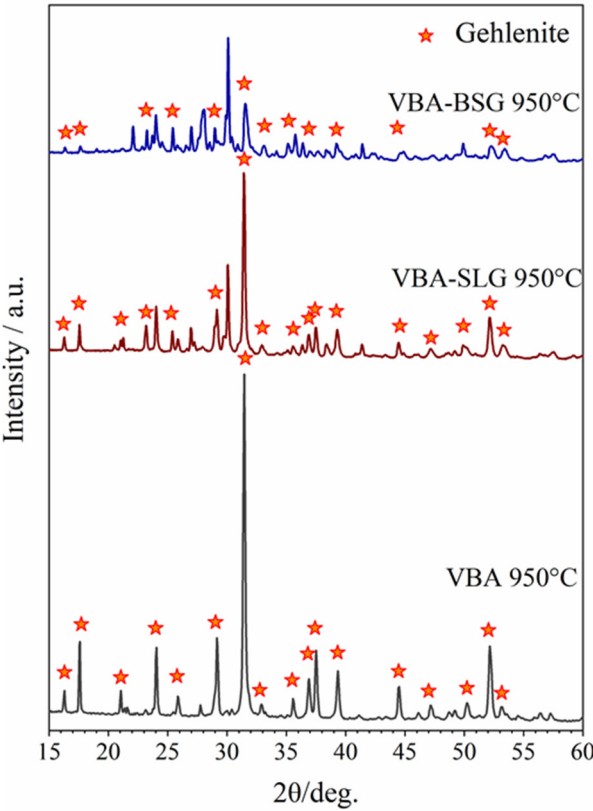

**Figure 3.** Decrease of gehlenite diffraction peaks, at 950 °C, caused by the addition of SLG and BSG glass.

The addition of SLG led to consolidation at temperatures of 900 °C and higher. However, as shown in Figure 4a, gehlenite was already formed at 850 °C, along with wollastonite ($CaSiO_3$, PDF#89-6458). The wollastonite peaks increased with increasing firing temperature. Moreover, above 900 °C, the formation of nepheline ($NaAlSiO_4$, PDF#76-1864) and some modification in the structure of gehlenite could not be excluded. Gehlenite belongs to melilite phases, which are known to form a wide range of solid solutions. The peaks at 850 °C were compatible with gehlenite, whereas the pattern at

950 °C had a better match with Cr-modified gehlenite ($Ca_2Al(Al(Si_{0.974}Cr_{0.026})O_7$, PDF#77-1147). $Cr^{3+}$ inclusions in the melilite crystals are common, and were previously investigated due to the possible enhancement of acoustic and nonlinear characteristics of this crystal group [25]. The formation of solid solutions was also detected in the case of wollastonite, since the peaks matched better with Fe-containing variant ($Ca_{2.87}Fe_{0.13}(SiO_3)_3$, PDF#83-2198), from 950 °C.

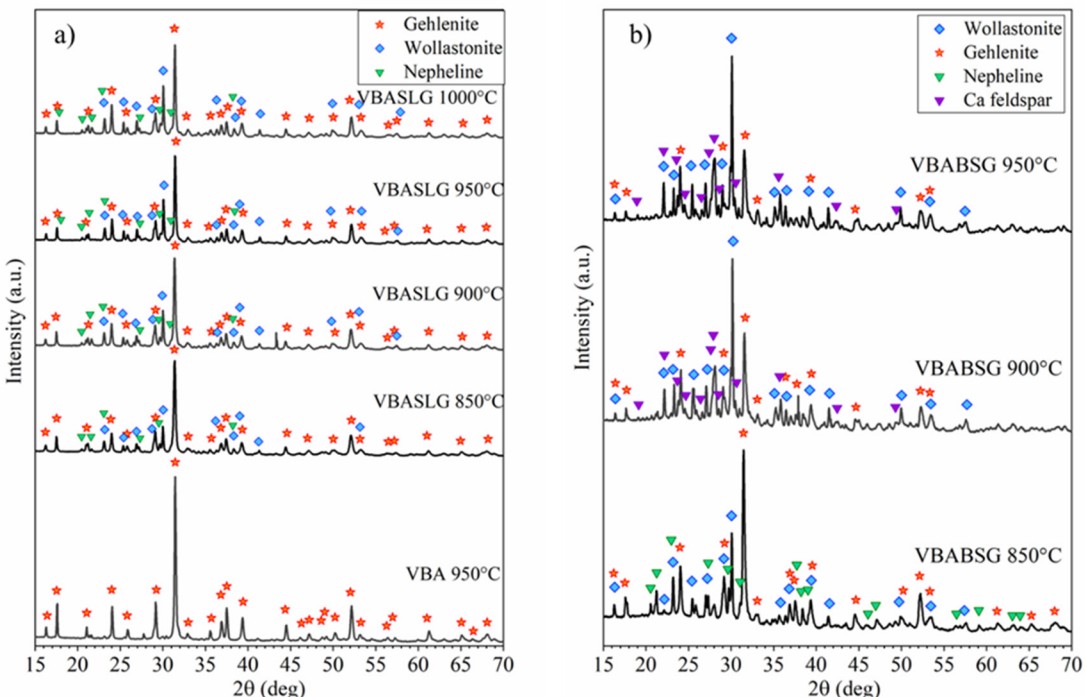

**Figure 4.** Mineralogical analysis of (**a**) VBA–SLG glass-ceramic foams and (**b**) VBA–BSG glass-ceramic foams.

The formation of wollastonite, as a transformation of gehlenite, is not surprising. Zhang and Liu [26], for example, produced wollastonite-based glass ceramics by mixing SLG with a gehlenite-based crystallization promoter (deriving from the firing of kaolin clay, $CaCO_3$, $BaCO_3$, and ZnO). VBA mixed with SLG likely acted similarly to the crystallization promoter, where at the same time the diffusion of aluminum ions in the glass may justify the formation of nepheline.

The curved background of the diffraction patterns of samples from VBA–BSG, shown in Figure 4b, suggests the presence of a residual amorphous phase. The VBA–BSG interaction was much more substantial than that of VBA–SLG, which was already expressed by the more intense wollastonite crystallization at 850 °C. More importantly, above 900 °C, wollastonite solid solution became the main phase. Interestingly, nepheline crystallized at 850 °C, but with increasing temperature it was replaced by labradorite (Ca-Na feldspar, $Ca_{0.64}Na_{0.35}(Al_{1.63}Si_{2.37}O_8)$, PDF#83-1371).

The more intensive interaction between VBA and BSG compared to that between VBA and SLG could be attributed to the different chemical formulation. BSG contains a higher content of $Al_2O_3$, a lower amount of $Na_2O$, and negligible CaO contribution. The transformation of gehlenite into anorthite (Ca feldspar) in the presence of silica and alumina ($Ca_2Al_2SiO_7 + 3SiO_2 + Al_2O_3 \rightarrow 2CaAl_2Si_2O_8$) is already known [24]. We suggest that the formation of nepheline and labradorite, as the products of VBA–glass reaction, replaced gehlenite, the standard product of VBA crystallization.

Figure 5 shows microstructural details of foams from VBA–SLG and VBA–BSG mixtures in the green state. In the case of SLG addition (Figure 5a), foams were less homogeneous, with coarser cells compared to those formed with BSG addition (Figure 5b). The morphology of samples did not change upon sintering, as an effect of the viscous flow freezing, caused by the crystallization (Figure 5c,d).

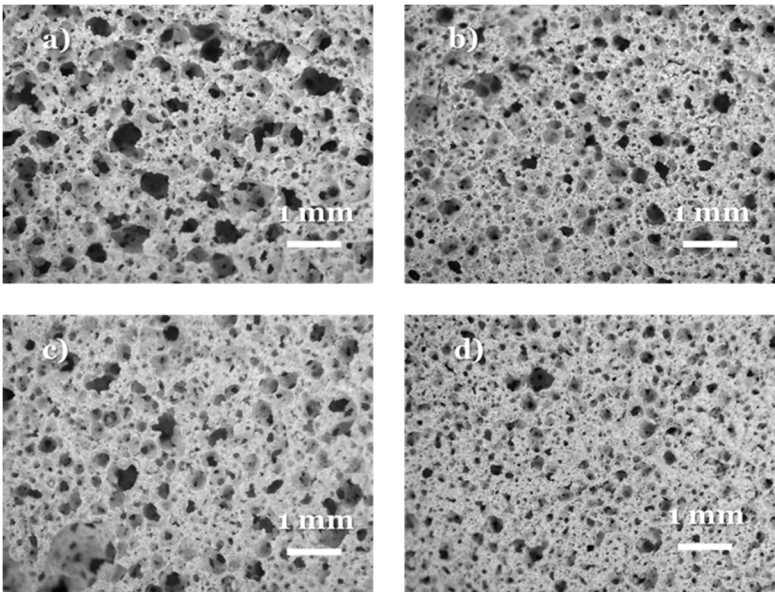

**Figure 5.** Micrographs of foams before and after firing at 950 °C: (**a,c**) VBA–SLG; (**b,d**) VBA–BSG.

Higher-magnification details for SLG addition are shown in Figure 6. The increasing firing temperatures did not cause any coarsening of the cellular structures (from 900 °C (Figure 6a) 950 °C (Figure 6c) and 1000 °C (Figure 6e)). This was due to the previously mentioned limitation of viscous flow caused by crystallization.

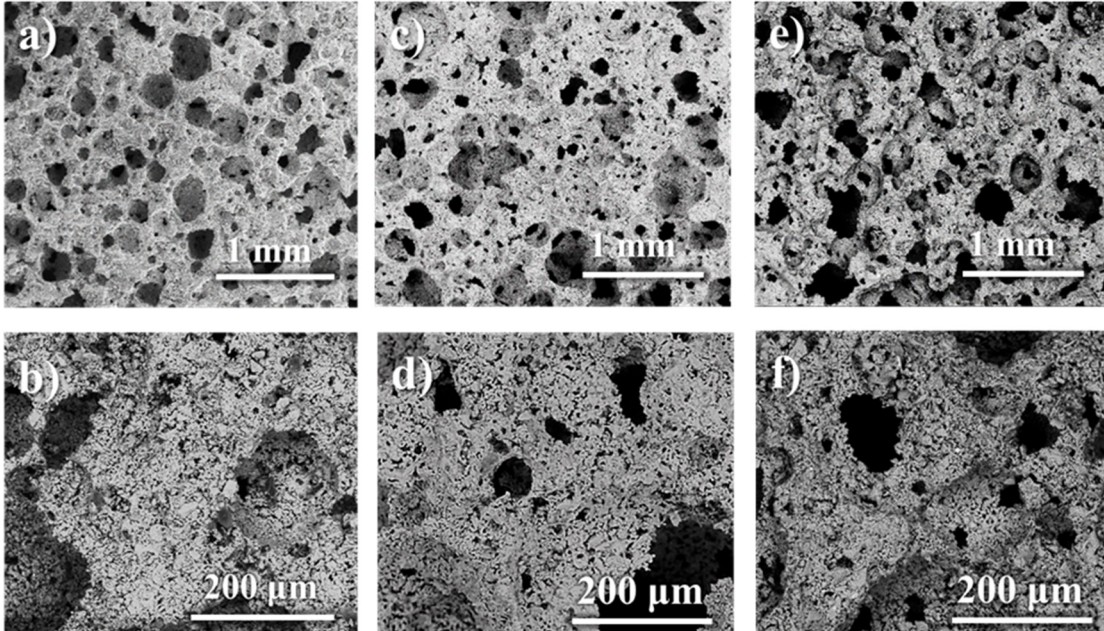

**Figure 6.** SEM pictures of VBA–SLG foams fired at 900 °C: (**a,b**), 950 °C (**c,d**), and 1000 °C (**e,f**).

In the case of BSG addition, the increase of firing temperature did not cause any coarsening (Figure 7a–f). Well-consolidated struts appeared at the temperatures starting from 900 °C (Figure 7c,d). The smooth areas indicate the presence of some residual glass phase.

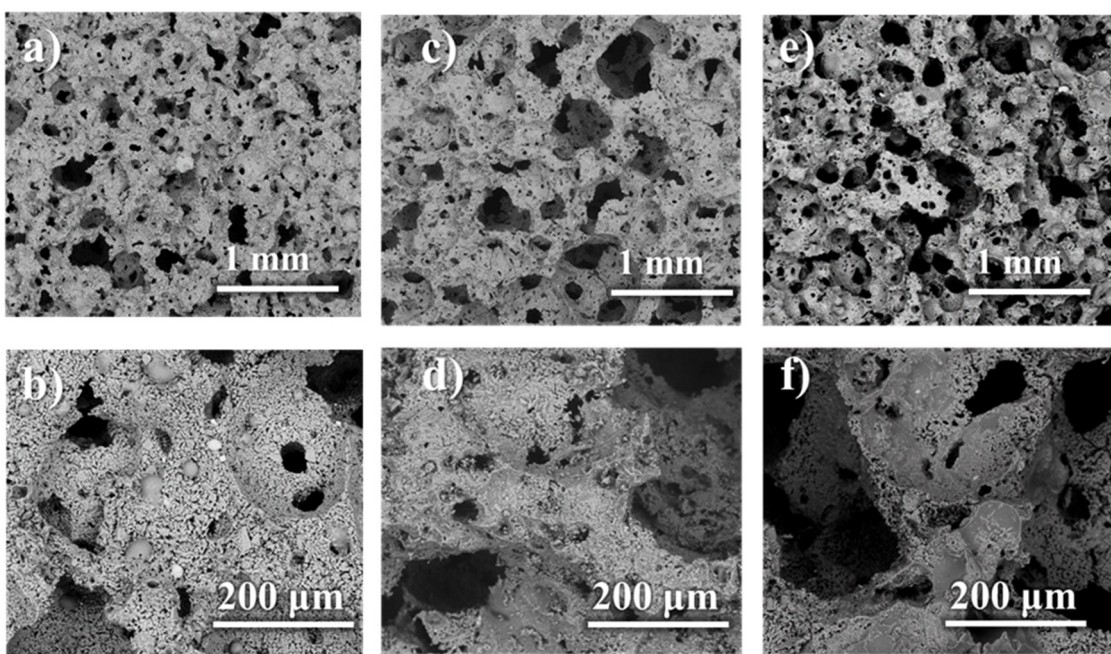

**Figure 7.** SEM pictures of VBA–BSG foams fired at 850 °C (**a,b**), 900 °C (**c,d**), and 950 °C (**e,f**).

Table 2 provides information on the mechanical properties and porosity of the prepared glass-ceramic foams. All foams were characterized by abundant porosity (total porosity >70 vol %), which was almost completely open. The strength/porosity correlation was studied via Gibson and Ashby's model [27], calculating the scaling of compressive strength with relative density for open-celled foams, according to the bending strength of the solid phase ($\sigma_{bend}$), as follows: $\sigma_{comp} = \sigma_{bend} \cdot [0.2 \cdot (\varrho_{rel})^{3/2}]$. From the experimental data of compressive strength and density, we could calculate the bending strength exceeding 100 MPa, comparing well with that of monolithic glass-ceramics of similar composition [23], for BSG addition. The addition of SLG did not lead to particularly strong foams.

**Table 2.** Values of porosity, as well as compressive and bending strength.

| Glass Additive/T | Porosity Total % | Open Porosity % | Closed Porosity % | Density g cm$^{-3}$ | Compressive Strength MPa | Bending Strength of the Solid Phase MPa |
|---|---|---|---|---|---|---|
| **SLG 900** | 76.9 | 75.2 | 1.7 | 0.68 (±0.03) | 1.7 (±0.4) | 77.1 |
| **SLG 950** | 73.5 | 74.9 | 0.1 | 0.70 (±0.01) | 2.0 (±0.2) | 71.4 |
| **SLG 1000** | 80.0 | 79.1 | 0.9 | 0.53 (±0.03) | 1.5 (±0.1) | 84.5 |
| **BSG 850** | 71.9 | 67.5 | 4.4 | 0.73 (±0.03) | 3.4 (±0.4) | 114.0 |
| **BSG 900** | 74.2 | 72.1 | 2.1 | 0.70 (±0.02) | 3.7 (±0.5) | 139.6 |
| **BSG 950** | 73.5 | 72.1 | 1.4 | 0.72 (±0.02) | 3.6 (±0.3) | 132.0 |

The much lower strength-to-density values for SLG could be justified by the above-mentioned different degree of interaction between VBA and glass type, which led to different phase assemblages, and internal stresses upon cooling from sintering temperature. In particular, melilites (including gehlenite) have much higher thermal expansion coefficients [28] than wollastonite and labradorite [29].

Table 3 shows data from leaching tests. All but the VBA–SLG fired at 900 °C met the requirements established in the limit values for waste acceptable at landfills for inert waste and nonhazardous waste [30]. In combination with the results from the mineralogical analysis, we can conclude that the increasing sintering temperatures affected the inclusion of chromium ions into gehlenite solid solutions. At lower temperature, $Cr^{3+}$ ions likely remained in the residual amorphous phase. This theory is also supported with higher $Cr^{3+}$ leaching from VBA–BSG sample fired at 850 °C, exhibiting a remarkable Cr release, although remaining well below the established threshold. In this case, the stabilization of the heavy metal benefited from the higher chemical durability of pharmaceutical glass.

**Table 3.** Results of the leaching tests performed on the VBA-based glass-ceramic foams. Concentrations are listed in ppm. Concentrations in bold exceeded the limiting values for inert waste.

| c [ppm] | Limit 1 * | Limit 2 ** | VBA AR *** | VBASLG7030 900 °C | VBASLG7030 950 °C | VBASLG7030 1000 °C | VBABSG7030 850 °C | VBABSG7030 900 °C | VBABSG7030 950 °C |
|---|---|---|---|---|---|---|---|---|---|
| As | 0.5 | 2 | <0.0049 | 0.0267 | 0.0132 | <0.0049 | <0.0049 | 0.0171 | 0.0102 |
| Ba | 20 | 100 | 0.0729 | 0.0082 | 0.0128 | 0.0438 | 0.0402 | 0.0105 | 0.0721 |
| Cd | 0.04 | 1 | 0.0003 | <0.0002 | <0.0002 | <0.0002 | <0.0002 | <0.0002 | <0.0002 |
| Cr | 0.5 | 10 | 0.0226 | **0.9596** | 0.1104 | 0.3079 | 0.2602 | 0.0409 | 0.0628 |
| Cu | 2 | 50 | 0.0902 | 0.0022 | 0.0100 | 0.0517 | 0.0620 | 0.0205 | 0.0642 |
| Mo | 0.5 | 10 | <0.0033 | <0.0033 | <0.0033 | <0.0033 | <0.0033 | <0.0033 | <0.0033 |
| Ni | 0.4 | 10 | <0.0014 | <0.0014 | <0.0014 | <0.0014 | <0.0014 | <0.0014 | <0.0014 |
| Pb | 0.5 | 10 | 0.0161 | <0.0047 | <0.0047 | 0.0182 | 0.0079 | <0.0047 | <0.0047 |
| Sb | 0.06 | 0.7 | <0.0099 | <0.0099 | <0.0099 | <0.0099 | <0.0099 | <0.0099 | <0.0099 |
| Se | 0.1 | 0.5 | 0.0525 | 0.0154 | 0.0124 | <0.0122 | <0.0122 | <0.0122 | <0.0122 |
| Zn | 4 | 50 | <0.0203 | <0.0203 | <0.0203 | <0.0203 | <0.0203 | <0.0203 | <0.0203 |

* According to the limit values for waste acceptable at landfills for inert waste [29]; ** According to the limit values for nonhazardous waste [29]; *** Refers to VBA as received in the form of powder <75 μm.

## 4. Conclusions

A relatively silica-poor VBA was successfully converted into highly porous glass-ceramic foams, by alkali activation and sinter crystallization. The addition of waste glasses, such as common soda-lime and pharmaceutical glasses, was essential in controlling the gelation of alkali-activated suspensions, which in turn enabled foaming by intensive mechanical stirring, and sintering. In fact, the two glasses enabled the consolidation of foamed suspensions at moderate temperatures and modified the crystallization, according to VBA–glass interactions. However, the effectiveness of reactive sintering had to be confronted with the requirements of chemical stability: the addition of pharmaceutical boro-alumino-silicate provided the best compromise between low processing temperature (850 °C), strength, and immobilization of heavy metals.

**Author Contributions:** M.H. (Miroslava Hujova) and P.R.M., preparation and analysis of samples; J.S. and M.H. (Miroslav Hnatko) project consultation; J.K. and E.B., writing and editing of the article; D.G., project supervision. All authors have read and agreed to the published version of the manuscript.

**Funding:** This paper is a part of the dissemination activities of the project FunGlass (Centre for Functional and Surface Functionalized Glass). This project has received funding from the European Union's Horizon 2020 research and innovation programme under grant agreement No 739566. Miroslava Hujova also gratefully acknowledges the financial support from Slovak Grant Agency of Ministry of Education, Science, Research and Sport, VEGA 2/0091/20 and from Slovak Research and Development Agency grant APVV-15-0540.

**Conflicts of Interest:** The authors declare no conflict of interest.

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
