# Peer review of "Glass-Ceramic Foams from Alkali-Activated Vitrified Bottom Ash and Waste Glasses"

_applsci, doi:10.3390/app10165714_

Round 1
Reviewer 1 Report
Table 1 shows the composition of Vitrified Bottom Ash, whether you checked whether this composition enters the glass formation zone in the system SiO2-CaO-Al2O3? Or is it glass or some other amorphous material not important for your research?
The DTA curve looks unusual (Fig 2). Do you have a non-linear baseline or a very weak signal? No axis labels? Therefore, it is impossible to estimate the magnitude of the crystallization peak. Also, the graph shows more than one crystallization peak.
What is the volume fraction of the crystalline phase in your glass ceramics?
Table 3 is not very easy to read. For example, the values for nickel, molybdenum, zinc and antimony do not depend on any parameters. Also, the dimension of the given values is not clear. Is this all ppm? Or just the first 2 columns?
Conclusions presented in fairly general terms.
Author Response
Dear Reviewer,
thank you kindly for reading and evaluating our manuscript. We have seriously considered each of your recommendations and modified the manuscript accordingly. Detailed replies to your observations are reported below.
Sincerely, (on behalf of all co-authors)
Miroslava Hujova
R. Reviewer
A. Authors
R. Table 1 shows the composition of Vitrified Bottom Ash, whether you checked whether this composition enters the glass formation zone in the system SiO2-CaO-Al2O3? Or is it glass or some other amorphous material not important for your research?
A. Glasses with similar composition were already presented in publications i.e. by Takahashi S. et.al, 2014, Palou M. et al, 2005 etc., now cited. Glasses of the system SiO2-CaO-Al2O3 resembling the composition of the (silica-poor) mineral gehlenite are known to be feasible, although processed at high temperatures. Vitrification at the reported conditions is assumed to be achieved also with the help of the (many) extra oxides.
R. The DTA curve looks unusual (Fig 2). Do you have a non-linear baseline or a very weak signal? No axis labels? Therefore, it is impossible to estimate the magnitude of the crystallization peak. Also, the graph shows more than one crystallization peak.
A. The reviewer is right, our equipment did not lead to a linear baseline (there was also a misunderstanding on the specific equipment adopted). We adjusted the Y scale (also including labels). We agree, there were several crystallization peaks. We focused on the first one, reasonably corresponding to the main crystal phase, i.e. gehlenite. We revised the text.
R. What is the volume fraction of the crystalline phase in your glass-ceramics?
A. We did not apply a quantitative analysis. The quantification of crystal phases, in waste-derived glass-ceramics, is always quite doubtful, since Rietveld refinements are affordable after the selection of ‘structural models’, regarding the position of all ions in the unit cells. The formation of solid solutions obviously complicates the identification of reference structural models. We simply relied on the outputs of the Match! Software package, concerning the identification of phases. The flat background of diffraction patterns, however, is a clear sign of abundant crystallization.
R. Table 3 is not very easy to read. For example, the values for nickel, molybdenum, zinc, and antimony do not depend on any parameters. Also, the dimension of the given values is not clear. Is this all ppm? Or just the first 2 columns?
A. Please see the revised version of the table in the manuscript. All values are in ppm. The mentioned elements were not detected, their concentration being well-below the sensitivity of our instrument.
R. Conclusions presented in fairly general terms.
A. We revised the conclusions.
Reviewer 2 Report
The authors have shown a relatively simple approach to utilize vitrified bottom ashes for the production of glass-ceramic foams which can be utilized for applications in thermal and acoustic insulation. Although similar strategies have already been performed by other authors, the authors have innovated by showing a two-sided approach, which reutilizes not only the vitrified bottom ashes, but also pharmaceutical and soda-lime glass waste. The manuscript is well written and I recommend for publication in Applied Sciences after minor revision regarding the following aspects:
- The novelty of this work should be better emphasized in the introduction. Although other authors have already synthesized glass-ceramic foams from VBA, the approach with waste glasses (SLG and BSG) is new.
- Table 1 – The authors should provide experimental errors for the fractions obtained from XRF. Are the concentrations for SLG and BSG nominal values or also measured by XRF? The values for SLG doesn’t sum up to 100. Is the missing 1,1% related to residues?
- I am not familiar with XRF technique and wonder if chromium species couldn’t be detected in the starting materials.
- Line 95 – 5 ºC∙min-1.
- IR results could be provided as supplemental material.
- Line 142 – The sentence is a bit confusing, please revise.
- DTA curve shows a second crystallization peak at ~1000 ºC. Was the crystallized phase identified?
Author Response
Dear Reviewer,
thank you kindly for reading and evaluating our manuscript. We have seriously considered each of your recommendations and modified the manuscript accordingly. Detailed replies to your observations are reported below.
Sincerely, (on the behalf of all co-authors)
Miroslava Hujova
R. Reviewer
A. Authors
R. The novelty of this work should be better emphasized in the introduction. Although other authors have already synthesized glass-ceramic foams from VBA, the approach with waste glasses (SLG and BSG) is new.
A. Previous works by listed authors were utilizing waste glasses. The novelty of this work is in dealing with VBA that is relatively silica-poor, reminding more of the amorphous gehlenite materials. This aspect is now better mirrored in the introduction of the manuscript.
R. Table 1 – The authors should provide experimental errors for the fractions obtained from XRF. Are the concentrations for SLG and BSG nominal values or also measured by XRF? The values for SLG doesn’t sum up to 100. Is the missing 1,1% related to residues?
A. Unfortunately, the experimental errors for the XRF values are not at our disposition. Chemical compositions for all of our materials were measured by the means of XRF. We would like to thank our reviewer: indeed, the residues are 1.1%. The elements that were present in less than 0.2% were omitted in this category for the brevity of information.
R. I am not familiar with the XRF technique and wonder if chromium species couldn’t be detected in the starting materials.
A. Chromium and other metals are part of the VBA. However, its amount is well-below 0.2% and therefore we decided to not list it in Table 1.
R. Line 95 – 5 ºC∙min-1.
A. Thank you for the observation, the text has been corrected.
R. IR results could be provided as supplemental material.
A. We have performed an FTIR study on our samples, we have uploaded a supplemental figure with an explanation.
R. Line 142 – The sentence is a bit confusing, please revise.
A. We have revised it. Hopefully, the sentence is now more concise and easier to follow.
R. DTA curve shows a second crystallization peak at ~1000 ºC. Was the crystallized phase identified?
A. Due to the specific composition of the VBA and its reminiscence of amorphous gehlenite materials, we have focused on the first crystallization peak that is assigned to the crystallization of the gehlenite phase. This is also due to the fact that our upcycling strategy employs sinter-crystallization, which is usually important within the window between glass transition temperature and first crystallization peak.
Reviewer 3 Report
The work investigates the conversion of vitrified bottom ash and waste glass mixtures (soda-lime glass or borosilicate glass) into highly porous glass-ceramic foams by the combination of alkali activation and sintering at 850-1000ͦC. A lot of characterization has been performed e.g. XRD, SEM, porosity, mechanical properties and ICP-OES analysis.
The present manuscript is interesting. The study is simple and suggests that a satisfactory compromise between low sintering temperature, promising mechanical property and better chemical resistance of the glass-ceramic foams can only be reached at certain specified conditions.
Page 2 , lines 78-79.
What is the vitrified temperature of the bottom ash using DC plasma reactor? what is the reason behind its melting before its combination with waste glasses? This treatment does incur more costs to the conversion of wastes into glass-ceramic foams.
It would be better to use melting instead of smelting. Melting converts a solid substance into a liquid whereas smelting converts an ore to its purest form.Smelting is the process by which a metal is extracted from its ore or mineral. For example, we would say, smelting of ores (e.g. iron titanium oxide, chromite, or nickel ore) in the presence of fluxes and reducing agent at more than the melting point of iron to get iron metal (pig iron) and titanium slag. The liquid melt at more than 1700 ͦC is then poured into molds to separate iron metal from titanium slag.
Page 3 , line 78-79. It would be better to reorganize these 2 lines. For example, I would suggest "Powdered VBA was suspended in 2.5M NaOH aqueous solution, either in its pure form or in combination with either SLG or BSG in a 70/30 weight ratio,.........
or any other modification it seems convenient to the authors
Page 3 , line 106 -107 . The number 10 g or 10 ml. DIW should be written de-Ionized water
Page 5 Figure 3. The peak at 2 theta 30 should be identified as wollastonite at VBA-SLG 950C and VBA-BSG at 950C
Page 5 , lines 164-167. In Table 1 (Chemical composition (in wt.%) of waste streams), there is no information about chromium. It should be included with other impurities in the 1.6% residue, Otherwise where it comes from?
Page 6, Figure 4. In comparison with SLG, the main peaks of gehlenite are decreasing gradually with increasing temperatures in the presence of BSG .What would be the situation of VBABSG at 1000ͦ C. The chart in not included
Page 7 Figure 5. Please check the figure caption
Author Response
Dear Reviewer,
thank you kindly for reading and evaluating our manuscript. We have seriously considered each of your recommendations and modified the manuscript accordingly. Detailed replies to your observations are reported below.
Sincerely, (on behalf of all co-authors)
Miroslava Hujova
R. Reviewer
A. Authors
R. The present manuscript is interesting. The study is simple and suggests that a satisfactory compromise between low sintering temperature, promising mechanical property and better chemical resistance of the glass-ceramic foams can only be reached at certain specified conditions.
A.We thank the reviewer for the observation. It well summarizes our approach.
R. Page 2 , lines 78-79. What is the vitrified temperature of the bottom ash using DC plasma reactor? what is the reason behind its melting before its combination with waste glasses? This treatment does incur more costs to the conversion of wastes into glass-ceramic foams. It would be better to use melting instead of smelting. Melting converts a solid substance into a liquid whereas smelting converts an ore to its purest form. Smelting is the process by which a metal is extracted from its ore or mineral. For example, we would say, smelting of ores (e.g. iron titanium oxide, chromite, or nickel ore) in the presence of fluxes and reducing agent at more than the melting point of iron to get iron metal (pig iron) and titanium slag. The liquid melt at more than 1700 ͦC is then poured into molds to separate iron metal from titanium slag.
A. The temperature of smelting has been added in the manuscript. The reason of the use of smelting technology instead of conventional melting was due to the fact that bottom ash consists of some harsh elements or compounds that are considered to be hazardous. During the smelting process, these are reduced in metallic form and thus are not present (or are present only in a limited amount) in a vitreous phase. Such extraction is therefore very advantageous in further improvement of vitreous phase leaching. Moreover, the energy consumption during the DC plasma treatment is much lower compared to conventional melting.
R. Page 3, line 78-79. It would be better to reorganize these 2 lines. For example, I would suggest "Powdered VBA was suspended in 2.5M NaOH aqueous solution, either in its pure form or in combination with either SLG or BSG in a 70/30 weight ratio,.........or any other modification it seems convenient to the authors
A. We thank the reviewer for the suggestion, the text was reorganized.
R. Page 3, line 106 -107 .The number 10 g or 10 ml. DIW should be written de-Ionized water
A. Thank you for the correction, we have changed it in the manuscript. The value is indeed 10 g.
R. Page 5 Figure 3.The peak at 2 theta 30 should be identified as wollastonite at VBA-SLG 950C and VBA-BSG at 950C
A. The reviewer is right, but with Fig.3 we just wanted to underline the dramatic change in the crystallization (of gehlenite) at 950°C operated by SLG and BSG. The crystal phases are discussed in detail with Fig.4a (SLG addition) and Fig.4b (BSG addition). We modified the caption as a reminder of our specific goal with Fig.3.
R. Page 5, lines 164-167. In Table 1(Chemical composition (in wt.%) of waste streams), there is no information about chromium. It should be included with other impurities in the 1.6% residue, Otherwise where it comes from?
A. Chromium is indeed included in the additional 1.6% of minor components. For the brevity of information, we decided to not list any component below 0,2% of wt.%
R. Page 6, Figure 4. In comparison with SLG, the main peaks of gehlenite are decreasing gradually with increasing temperatures in the presence of BSG. What would be the situation of VBABSG at 1000 °C. The chart in not included
A. Unfortunately, the sample VBABSG at 1000°C was not prepared: the main aim of the research was to find the most ideal combination between composition/temperature/mechanical properties/chemical durability of our materials. In the case of the BSG line of samples, it is shown that growing temperatures are not favorable for this composition. Therefore, we have not included VBABSG 1000°C into the study.
R. Page 7 Figure 5. Please check the figure caption
A. We have corrected the caption. Thank you for your observation.
Reviewer 4 Report
This paper deals with a new technology to produce glass-ceramic foams with alkali-activation. The topic is interesting and the English writing of this paper is good. But I have some doubts. I suggest a major revision, not meaning a very big change of the paper, but I need to see more information and explanations, and then judge whether it can be accepted.
Line 47. Based on which reference/report? How do you know this?
Does VBA contain metallic Al? Al is also a commonly used foaming agent. I know it will dissolve during the mixing with NaOH solution for 2h, but I want to know how much the H2 released from the Al contribute to the foam? In this study you didn’t give the Al content, only in XRF in the form of Al2O3. You know XRF cannot distinguish Al and Al2O3. So I hope the authors can add this information.
In addition, why you choose 2h of mixing? Any reference, or what is the consideration about the duration?
You mentioned limited gel is formed, so the alkali activation actually doesn’t contribute much? Then why don’t you use water, can the foam still form then? If not to the gel, what is the exact contribution of the alkali? Maybe just to increase the viscosity?? Please explain.
In figure 1, there is quartz in the as-received ash. Then after activation, it disappears. The NaOH solution can dissolve quartz? I need explanation.
Author Response
Dear Reviewer,
thank you kindly for reading and evaluating our manuscript. We have seriously considered each of your recommendations and modified the manuscript accordingly. Detailed replies to your observations are reported below.
Sincerely, (on behalf of all coauthors)
Miroslava Hujova
R. This paper deals with a new technology to produce glass-ceramic foams with alkali-activation. The topic is interesting and the English writing of this paper is good. But I have some doubts. I suggest a major revision, not meaning a very big change of the paper, but I need to see more information and explanations, and then judge whether it can be accepted.
A. We understand the approach. We hope that the additional information provided with the new version will be helpful.
R. Line 47. Based on which reference/report? How do you know this?
A. The line 47 is information coming from report that is in line 48 addressed by number 11. This exceptional report by Blasenbauer and collective was released earlier this year and contains very useful information on VBA management, some of which we paraphrased in our manuscript. However, we understand your confusion and for the better clarity we duplicated the reference number for the convenience of reader to prevent any confusion.
R. Does VBA contain metallic Al? Al is also a commonly used foaming agent. I know it will dissolve during the mixing with NaOH solution for 2h, but I want to know how much the H2 released from the Al contribute to the foam? In this study you didn’t give the Al content, only in XRF in the form of Al2O3. You know XRF cannot distinguish Al and Al2O3. So I hope the authors can add this information.
A. The metal can be excluded owing to the processing conditions through which the VBA is obtained. Furthermore, as illustrated in the diffraction pattern of VBA, no crystalline Al could be detected. Al has a distinct main peak (1 1 1), centred at 39°, not visible. Since Al is not present we cannot address evolution of hydrogen as the result of reaction between Al and NaOH
R. In addition, why you choose 2h of mixing? Any reference, or what is the consideration about the duration?
A. The duration of mixing was chosen according to previous studies (by Rincon Romero, Rabello Monich and Elsayed), mentioned in the manuscript. Especially article by Rincon Romero from 2018 “Up-cycling of vitrified bottom ash from MSWI into glass-ceramic foams by means of ‘ inorganic gel casting ’ and sinter-crystallization“ (Ref 17) was used as the proper strategy for our study. For better clarity, we added this citation also to the references in the Experimental part of the manuscript.
R. You mentioned limited gel is formed, so the alkali activation actually doesn’t contribute much? Then why don’t you use water, can the foam still form then? If not to the gel, what is the exact contribution of the alkali? Maybe just to increase the viscosity?? Please explain.
A. The reviewer is right. The gel formed by alkali activation essentially modifies the viscosity of the suspensions. As done in previous studies, we were not interested at extensive dissolution of glasses (VBA, SLG and BSG), according to the low molarity of NaOH. The formation of a (semi-crystalline) gel was intended to ‘freeze’ the cellular structure, obtained after intensive mechanical stirring, during drying.
The formation of surface gels is typical of an alkaline attack of glass. OH- anions disrupt the glass network, by breaking Si-O-Si bonds, but a complete dissolution is often hindered by the formation of a ‘passivating layer’ of gel from corrosion products, with variable solubility according to the glass chemistry (as an example, calcium silicate hydrate, C–S–H, gel layers are quite protective [https://ceramics.onlinelibrary.wiley.com/doi/full/10.1111/jace.14933]). Pure water is known to attack glass by a ‘mixed’ mechanism, comprising the acid leaching of some modifying ions (e.g. Na+ ions from glass exchanged with H+ from water), followed by basic attack (operated by OH- ions from water, no longer balanced by H+ ions entered in the glass structure). The attack may be more or less significant, again depending on composition (as an example, glasses fully corresponding to the stoichiometry of the mineral gehlenite – as reported by Palou M. et al, 2005, now cited - are soluble in water and form a cementitious paste). VBA, owing to its particular composition, did not exhibit any significant reaction with pure water; the reduction of alkaline activator, for the sake of sustainability, anyway, will undoubtedly constitute the focus of future investigations.
R. In figure 1, there is quartz in the as-received ash. Then after activation, it disappears. The NaOH solution can dissolve quartz? I need explanation.
A. We cannot exclude some dissolution of quartz impurity in the SLG. Anyway, alkali-activated materials in Fig.1 (top patterns) represent mixtures based on VBA (SLG and its impurities were somewhat ‘diluted’).
Round 2
Reviewer 4 Report
thanks for the reply. So the metal Al does not exist, which is because the high temperature (1400 as you mentioned in the reply to other reviewers) during the production of the ash, right? Then I understand.
I didn't mean to ask you to reduce the Naoh concentration in future study. I just want the author to add some words on the influence of alkalis in the paper as they said in the reply: although there is not much gel formed, the alkalis has other contributions like to ‘freeze’ the cellular structure, obtained after intensive mechanical stirring, during drying.
No other comments. Can be accepted after addressing the above 2 very minor questions.
Author Response
R. Reviewer
A. Authors
R. thanks for the reply. So the metal Al does not exist, which is because the high temperature (1400 as you mentioned in the reply to other reviewers) during the production of the ash, right? Then I understand.
A. The reviewer is right. The presentation of VBA in the previous submission had some mistakes. We modified the manuscript to underline that treatments on bottom ash are specifically determine the recovery of metals, so that the residue (VBA) is metal-free.
R. I didn't mean to ask you to reduce the Naoh concentration in future study. I just want the author to add some words on the influence of alkalis in the paper as they said in the reply: although there is not much gel formed, the alkalis has other contributions like to ‘freeze’ the cellular structure, obtained after intensive mechanical stirring, during drying.
A. We modified the manuscript to clarify that our concept is of a ‘weak’ alkaline activation. The ‘freezing’ of cellular structure, developed upon frothing, is due to the viscosity increase, in turn motivated by the formation of gels.